# The Simulated Physiological Oocyte Maturation (SPOM) System Enhances Cytoplasmic Maturation and Oocyte Competence in Cattle

**DOI:** 10.3390/ani14131893

**Published:** 2024-06-27

**Authors:** Micaela Navarro, Tomás Fanti, Nicolas Matias Ortega, Magalí Waremkraut, Francisco Guaimas, Adrian Ángel Mutto, Carolina Blüguermann

**Affiliations:** Instituto de Investigaciones Biotecnológicas ‘Dr Rodolfo Ugalde’ (IIBIO), UNSAM-CONICET, Buenos Aires CP 1650, Argentina; minavarro@iib.unsam.edu.ar (M.N.); tomi.fanti@gmail.com (T.F.); nicolasortega.biomol@gmail.com (N.M.O.); magaliwarem@gmail.com (M.W.); fguaimas@iib.unsam.edu.ar (F.G.); cbluguermann@iib.unsam.edu.ar (C.B.)

**Keywords:** bovine, in vitro maturation, oocyte competence, cytoplasmic maturation, cAMP modulators, embryo development, mitochondrial activity

## Abstract

**Simple Summary:**

In vitro embryo production in cattle still needs improvement in order to reach quality and pregnancy rates comparable to in vivo-derived embryos. One of the limitations of this technique is related to the poor in vitro maturation of oocytes, mainly related to incomplete cytoplasmatic maturation. In the present work, we studied the effect of regulating cytoplasmatic maturation on the blastocyst production rate and quality. We found out that oocytes that were cultured in the presence of cyclic adenosine monophosphate modulators reached a maximum of mitochondrial activity earlier than not-treated oocytes, which led to a significantly improved blastocyst quality and production rate.

**Abstract:**

In vitro embryo production is a widely applied technique that allows the expansion of genetics and accelerated breeding programs. However, in cattle, this technique still needs improvement in order to reach quality and pregnancy rates comparable to in vivo-derived embryos. One of the limitations of this technique is related to in vitro maturation, where a heterogeneous population of oocytes is harvested from follicles and cultured in vitro in the presence of gonadotropic hormones to induce maturation. As a result, oocytes with different degrees of competence are obtained, resulting in a decrease in the quality and quantity of embryos obtained. A novel system based on the use of cyclic adenosine monophosphate (cAMP) modulators was developed to enhance bovine oocyte competence, although controversial results were obtained depending on the in vitro embryo production (IVP) system used in each laboratory. Thus, in the present work, we employed a reported cAMP protocol named Simulated Physiological Oocyte Maturation (SPOM) under our IVP system and analysed its effect on cytoplasmic maturation by measuring levels of stress-related genes and evaluating the activity and distribution of mitochondria as a marker for cytoplasmic maturation Moreover, we studied the effect of the cAMP treatment on nuclear maturation, cleavage, and blastocyst formation. Finally, we assessed the embryo quality by determining the hatching rates, total cell number per blastocyst, cryopreservation tolerance, and embryo implantation. We found that maturing oocytes in the presence of cAMP modulators did not affect nuclear maturation, although they changed the dynamic pattern of mitochondrial activity along maturation. Additionally, we found that oocytes subjected to cAMP modulators significantly improved blastocyst formation (15.5% vs. 22.2%, *p* < 0.05). Blastocysts derived from cAMP-treated oocytes did not improve cryopreservation tolerance but showed an increased hatching rate, a higher total cell number per blastocyst and, when transferred to hormonally synchronised recipients, produced pregnancies. These results reflect that the use of cAMP modulators during IVM results in competent oocytes that, after fertilisation, can develop in more blastocysts with a better quality than standard IVM conditions.

## 1. Introduction

In vitro reproductive techniques aim to increase mammalian reproduction efficiency. In cattle, IVP is constantly evolving in order to achieve a higher quantity and better quality of embryos. Apart from its important utility in developmental biology studies, this technique has a significant role in improving breeding programmes, producing embryos via somatic cell nuclear transfer, stem cell derivation, and producing transgenic animals, among others [1]. The IVP process consists of three subsequent events: maturation of the female gamete, sperm capacitation followed by fertilisation of matured eggs, and ultimately, the culture of presumptive zygotes until the blastocyst or expanded blastocyst stage. Despite the numerous applications of IVP, it is still a sub-efficient technique compared to in vivo embryo production [2].

Several factors can affect IVP at different steps of the procedure that would end in a lower embryo competence. Some examples include temperature and pH variations, unbalanced culture conditions, prolonged incubation times, and maternal ageing, among others. However, one of the main obstacles in cattle IVP is mimicking the processes that occur during in vivo oocyte maturation, which involves nuclear and cytoplasmic changes. In the initial stages of oocyte maturation, oocytes are meiotically arrested in prophase I due to the interaction with different factors that are present in the follicle that prevents activation of the maturation-promoting factor (MPF). cAMP plays a pivotal role in maintaining the meiotic arrest until cytoplasmic maturation is completed. The cAMP concentration is finely regulated by the follicular fluid and cumulus cells surrounding the oocytes [3]. The follicular fluid contains different molecules that modulate cAMP levels [4], while cumulus cells supply cyclic guanosine monophosphate to the oocyte, which inhibits phosphodiesterase (PDE) activity, the enzyme that degrades cAMP [5]. Adequate intra-oocyte concentrations of cAMP are required for final oocyte differentiation, such as chromatin transition and the gradual silencing of transcription [6].

While the progression of meiosis characterises nuclear oocyte maturation, it does not by itself ensure further embryonic development; additional cytoplasmic maturation is required, which includes maternal protein and RNA storage [7], the development of calcium regulatory mechanisms, changes in MPF activity, dynamic reorganisation of cytoskeletal filaments [8], and the redistribution of cortical granules and cellular organelles such as mitochondria [9]. During cytoplasmic maturation, mitochondria activity and distribution are modified: active mitochondria migrate and disperse from the peripheral regions towards the centre of the oocyte [10,11,12,13,14]. This phenomenon has been previously correlated with successful embryo development [15]. By the end of maturation, followed by a gonadotropic hormone signal, synchronised nuclear and cytoplasmic maturation will occur, resulting in a gamete which has resumed meiosis until the stage of metaphase II and has the competence to be fertilised and sustain initial embryo development until embryonic genome activation [16,17].

In the case of in vitro maturation (IVM), a heterogeneous population of oocytes is harvested from follicles at different stages of development and cultured in the presence of gonadotropic hormones to generate mature oocytes [18]. The mechanical removal of cumulus cells and follicular fluid produces an abrupt decrease in intracellular cAMP concentrations, generating precocious meiotic resumption and an incomplete cytoplasmic maturation, resulting in oocytes with varying degrees of competence [19]. This process, widely known as spontaneous nuclear maturation, is likely to occur in standard IVM protocols, where there is a lack of maternal signals from the follicle and cumulus cells that are required to coordinate the mechanisms of nuclear and cytoplasmic maturation that would guarantee developmental competence of the oocyte [17,20].

Currently, our lab and others have attempted to improve the IVM protocol to increase IVP rates and embryo quality [21,22]. Several researchers demonstrated that the modulation of the cAMP concentration during IVM could improve oocyte competence and described a novel ex vivo simulated physiological oocyte maturation system named SPOM, which consisted of a two-step maturation protocol that included the use of the cAMP modulators forskolin, 3-isobutyl-1-methylxanthine, and cilostamide [23,24,25]. Blocking meiosis during a pre-maturation step right after removing oocytes from the follicles has been suggested as an alternative to provide the oocyte with additional time to synchronise nuclear and cytoplasmic maturation [26]. This strategy has been evaluated in different species, such as humans, bovines, ovines, and caprines, among others, with varying degrees of success, making it hard to conclude if this modified IVM system can be generally implemented [27,28,29]. Additionally, there are no reports that show a direct effect of the SPOM system on cytoplasmic maturation. Thus, in the present study, we evaluated whether the SPOM protocol could improve oocyte competence in our IVP system and evaluated cytoplasmic maturation through the determination of stress-related gene levels and via the evaluation of mitochondria activity and distribution at critical time points during maturation.

## 2. Materials and Methods

### 2.1. Oocyte Collection and In Vitro Maturation

Bovine ovaries were collected from a local slaughterhouse in compliance with regulations for animal testing and research at the IIBIO-UNSAM (National Regulation Number 18819-1970). Cumulus–oocyte complexes (COCs) were recovered by aspirating follicles of 2–8 mm in diameter using 21-G needles. The follicular fluid (that contains the COCs) was collected in 50 mL tubes. While aspirating, the tube was placed in a water bath at 37 °C in a vertical position to allow COC precipitation. COCs were recovered from the bottom of the tube and washed twice in 35 mm plates containing 3 mL of washing media that contained Tissue Culture Medium (TCM) 199 with Hank’s salts (ThermoFisher, Waltham, MA, USA) supplemented with 3 mg/mL of bovine serum albumin (BSA), 10% foetal bovine serum (FBS, ThermoFisher, Waltham, MA, USA) and 100 µg/mL of gentamicin (ThermoFisher, Waltham, MA, USA). Only COCs with homogeneous cytoplasm and at least three layers of cumulus cells were selected for IVM. Standard IVM conditions consisted of incubating groups of 40–50 COCs in 4-well plates in the presence of 500 μL IVM base media (TCM199 with Earle’s salts supplemented with 1.1 mg/mL of sodium pyruvate (Sigma, Burlington, MA, USA), 100 µg/mL of gentamicin, 1 mg/mL of glutamine (Sigma), and 10% FBS) supplemented with 0.1 IU of recombinant human follicle-stimulating hormone (rhFSH, Organon, Jersey City, NJ, USA) for 22 h at 38.5 °C and 5% CO_2_ [30]. IVM in the presence of cAMP modulators was performed by incubating groups of 40–50 COCs in 4-well plates in IVM base media supplemented with 500 μM of 3-isobutyl-methylxanthine (IBMX, Sigma, Burlington, MA, USA) and 100 μM of forskolin (FSK, Sigma, Burlington, MA, USA) for 2 h at 38.5 °C and 5% CO_2_. Thereafter, COCs were thoroughly washed and transferred to IVM base media supplemented with 0.1 IU of rhFSH and 20 μM of cilostamide (CIL, Sigma, Burlington, MA, USA) and cultured for 22 h at 38.5 °C and 5% CO_2_. An illustration of the experimental design is exhibited in Figure 1.

To determine nuclear maturation, subsets of at least 50 COCs from each experimental condition were incubated for 4 min in 1 mg/mL of hyaluronidase solution (Sigma), and the presence of the first polar body was assessed using a stereomicroscope.

### 2.2. In Vitro Fertilisation (IVF) and In Vitro Embryo Culture (IVC)

Expanded COCs and frozen–thawed semen from Aberdeen Angus bulls were used for in vitro fertilisation. Groups of 40–50 expanded COCs were transferred to IVF-SOF media that consisted of 6.3 mg/mL of NaCl, 536 µg/mL of KCl, 40 µg/mL of KH_2_PO_4_, 2.1 mg/mL of NaHCO_3_, 65 µg/mL of penicillin, 55 µg/mL of sodium pyruvate, 8 mg/mL of BSA, 90 µg/mL of fructose, 252 µg/mL of CaCl_2_-2H_2_O, 0.23 µL/mL of sodium lactate (60%), and 50 µg/mL of heparin. Frozen semen was thawed by immersing straws for 30 s in a 37 °C water bath, followed by Percoll gradient (90%/45%) and centrifugation at 600× *g* for 15 min. Sperm were washed in H-TALP medium (5.8 mg/mL of NaCl, 40 µg/mL of NaH_2_PO_4_, 4 µL/mL of sodium lactate (60%), 2.19 mg/mL of NaHCO_3_, 384 µg/mL of CaCl_2_-2H_2_O, 310 µg/mL of MgCl_2_-6H_2_O, 6 mg/mL of BSA, and 2.39 mg/mL of Hepes) and centrifuged at 200× *g* for 5 min. Sperm concentration was determined using a Neubauer chamber. An insemination dose of 2 × 10^6^ sperm/mL was used per group of 40–50 COCs. Co-incubation of both gametes was performed at 38.5 °C and 5% CO_2_ for 20–22 h.

After IVF, presumptive zygotes were denuded by gently pipetting, and groups of 40–50 were cultured in Continuous Single Culture medium (CSC, Irvine, Santa Ana, CA, USA) supplemented with 6 mg/mL of BSA covered with mineral oil (Sigma) for 7–8 days at 38.5 °C, 5% CO_2_, and 5% O_2_. Cleavage rate was evaluated by determining the quantity of 2- or 4-cell embryos at 48 h post-fertilisation. Blastocyst and hatched blastocyst rates were studied after 7 or 8 days of IVC, respectively. Blastocysts from one session of IVP that were at day 7-IVC were used for embryo transfer (ET). ET was performed on hormonally synchronised recipients. For synchronisation, the intravaginal DIB 0.5 g device (Zoetis, Parsipanny, NJ, USA) was put on recipients together with 2 cc of estradiol benzoate. Nine days later, DIB was removed, and recipients were treated with 2 cc prostaglandin. On the next day, recipients were treated with 1 cc of estradiol benzoate, and estrus was detected up to 48 h later (day 0). On day 7 after estrus detection, embryo transfer was performed, where only recipients with a notorious corpus luteum were used. Two blastocysts at day 7-IVC per condition were loaded into 0.25 mL straws with commercial holding medium (Vetoquinol, Fort Worth, TX, USA) and transferred to the uterine horn of each recipient cow via a transcervical method. Pregnancy rate was determined by ultrasound at 60 days after ET.

### 2.3. Quantification of Total Cell Number

To determine total cell number of embryos, blastocysts at day 7-IVC were fixed using 4% paraformaldehyde solution for 30 min at room temperature and dyed with 5 μg/mL Hoechst 33,342 for 10 min. Stained embryos were observed under a fluorescence microscope, and images were taken using a Nikon Digital Sight camera and NIS-Elements BR 3.2 64-bit software. Total cell number was counted using Image J software (version 1.53a).

### 2.4. Evaluation of Cryopreservation Tolerance

Blastocysts at day 7-IVC were cryopreserved via conventional slow-freezing protocols. Briefly, embryos were incubated with 1.5 M of ethyleneglycol (Riedel-deHaen, Seelze, Germany), 0.1 M of sucrose, and 10% FBS for 4 min at room temperature and loaded into 0.25 mL straws. Straws were transferred into a controlled-rate freezer for 10 min at −6.5 °C, followed by a cooling curve of −0.6 °C/min from −6.5 to −35 °C and then plunged into liquid nitrogen for storage.

To evaluate cryopreservation tolerance, frozen embryos were thawed by warming the straws for 10 s at room temperature, followed by 10 s at 37 °C and incubated with 0.5 M of sucrose and 10% FBS for 5 min at 38.5 °C. Afterwards, embryos were cultured in CSC medium supplemented with 6 mg/mL of BSA and 10% FBS covered with mineral oil at 38.5 °C, 5% CO_2_, and 5% O_2_ for 24 h. After 24 h, embryonic survival rate was assessed by observing blastocyst expansion or hatching using a stereomicroscope.

### 2.5. Determination of Relative mRNA Abundance of Stress-Related Genes

Relative abundance of stress-related genes was determined in mature oocytes using real-time PCR (qPCR). Total RNA was isolated from groups of 10 oocytes from each IVM condition (standard or in the presence of cAMP modulators) using TRIzol reagent following manufacturer’s instructions with a brief modification. To increase total RNA concentration, 3.5 µg of yeast tRNA (ThermoFisher, Waltham, MA, USA) and 20 µg of glycogen (ThermoFisher, Waltham, MA, USA) were added as carriers. After RNA isolation, removal of residual genomic DNA was performed using DNA-free^TM^ kit (Ambion, Austin, TX, USA) according to the manufacturer’s instructions. The whole eluted (~10 µL)was subjected to reverse transcription. Reverse transcription reaction was performed using SuperScript^TM^ II Reverse Transcriptase (ThermoFisher, Waltham, MA, USA) following manufacturer’s guidelines. Gene expression was quantified by qPCR using SYBR Green PCR Master Mix (Kappa Biosystems, Wilmington, DE, USA) in Applied Biosystems 7500 Real-Time PCR System (Applied Biosystems, Foster City, CA, USA) according to the manufacturer’s protocol. Oligonucleotides used for this study are listed in Table 1. Three independent samples were run in technical triplicates, and relative expression was determined using the standard curve method. Standard curves were created by serial dilutions (1/2) of pooled cDNA samples from both IVM conditions. Data were normalised to *hmbs* housekeeping gene since the expression of this gene had a stable pattern between samples and conditions.

### 2.6. Analysis of Mitochondrial Activity and Distribution

Groups of 10 oocytes at 0, 7, 15, and 22 h of maturation (setting time of 0 h when rhFSH was added to the maturation medium) were denuded and stained without fixation to evaluate mitochondrial activity and distribution. Active oocyte’s mitochondria were stained using 100 nM of Mitotracker Red CMXROS (Molecular Probes, Eugene, OR, USA) prepared in Dulbecco’s Phosphate-Buffered Saline (DPBS) supplemented with 3 mg/mL of BSA for 30 min at 38.5 °C. After staining, oocytes were washed in 5 drops of 100 µL of washing media and then mounted in a microscope slide using DPBS. Stained oocytes were examined using a laser scanning confocal microscope Olympus FV1000 (Olympus, Tokyo, Japan) equipped with a He-Ne 543 nm laser and FV10-ASW software (Soft Fluoview 3.1a). Laser power, 60× Oil PLAPON 1.42 NA, pinhole aperture, filter set, and voltage of the PMT were set at the beginning of the experiment and kept throughout the whole experiment. Oocytes were scanned, and photographs were taken every 3 µm (stack sizes ranged from 30 to 33 µm). For analysis, we made a projection of the summarised Z stack in one plane. The fluorescence intensity was measured in a selected donut-shaped area for each oocyte (the area was maintained the same for each measurement; see Figure 2 for a representative image of the measurement method) using Image J software (version number 1.53a). Results are expressed as fluorescence intensity over area.

### 2.7. Experimental Design and Statistical Analysis

We used a completely randomised experimental design. For this, once the total oocytes were collected, they were divided equally (or as similarly as possible) between the Standard IVM or SPOM IVM groups. *n* = indicates the number of times such an experiment was carried out. Each experiment included between 56 and 260 oocytes, depending on material availability. We performed a normality test (Shapiro–Wilks) for all the data sets presented in this work. Depending on the data set, statistical analysis was performed using paired Student’s *t*-test for data set with normal distribution, Wilcoxon test, or two-way ANOVA with Bonferroni’s multiple comparisons test when data distribution was not normal. Values of *p* < 0.05 were considered significant and are shown with * or different letters.

## 3. Results

### 3.1. Effects of cAMP Modulators on Nuclear and Cytoplasmic Maturation

We began by determining if the presence of cAMP modulators during IVM has some effect on oocyte nuclear maturation. To that end, nuclear maturation was analysed through meiosis resumption by detecting the first polar body in the perivitelline space in oocytes matured in standard IVM conditions or in the presence of cAMP modulators. We did not find any differences in nuclear maturation in either IVM conditions (Figure 3), reflecting that the incubation with cAMP modulators did not alter nuclear maturation rates.

Our next step consisted of investigating the effects of cAMP modulators on cytoplasmic maturation. For this, we first evaluated the expression levels of the stress-related genes *bax*, *bip*, *psmβ5*, *atf6,* and *hsp70*, genes that are involved in apoptosis and protein folding processes and have been proposed as indicators of oocyte quality [30]. When comparing the relative amount of transcripts present in matured oocytes that were subjected to standard IVM (control) or IVM in the presence of cAMP modulators, no significant differences were found between both groups, suggesting that the presence of cAMP modulators does not modify the expression levels of the analysed genes and therefore does not seem to have an effect of cellular stress. However, it would be interesting to analyse a broader selection of stress-related genes to further determine the effects of cAMP on oocyte cytoplasmic maturation (Figure 4).

Since mitochondrial activity and distribution are known to be a crucial process during oocyte maturation and are directly related to oocyte competence [31], we sought to analyse the behaviour of this organelle in oocytes matured in the presence of cAMP modulators. Mitochondrial activity and distribution were evaluated at critical time points of maturation: 0, 7, 15, and 22 h in control and cAMP-treated oocytes. We observed that along maturation progression, mitochondria retained their peripheral distribution (Figure 5A); however, interesting differences were found in mitochondrial activity between both groups. cAMP-treated oocytes reached a maximum mitochondrial activity (evidenced by a higher fluorescence intensity) earlier than their control counterparts (7 h vs. 22 h), demonstrating a different pattern in mitochondrial dynamics during oocyte maturation between both groups (Figure 5B).

Altogether, these results suggest that the addition of cAMP modulators during the IVM procedure does not affect nuclear maturation or the expression of stress-related genes but enables a faster accumulation of active mitochondria that could have an impact on oocyte competence and future embryo development and quality.

### 3.2. The Use of cAMP Modulators during IVM Increases Bovine Embryo Development

Performing IVM in the presence of cAMP modulators has shown controversial results in bovine species, particularly in cleavage rates and blastocyst formation [1,32]. Thus, we continued our study by analysing cleavage and blastocyst development rates using our IVP system with an IVM step with or without cAMP modulators.

When comparing embryos derived from control or cAMP-treated oocytes, we did not observe statistical differences in the cleavage rate (83.74% and 83.76% for control and cAMP-oocytes, respectively) (Figure 3). However, when we evaluated blastocyst formation, we observed that a significantly higher number of embryos reached the blastocyst stage when they were derived from cAMP-oocytes compared to control-derived oocytes (15.5% vs. 22.2%, respectively, *p* < 0.05) (Figure 3). These results suggest that the presence of cAMP modulators during the IVM step did not change the cleavage rate but resulted in an increased rate of blastocyst production.

### 3.3. Oocytes Matured in the Presence of cAMP Modulators Show an Enhanced Embryo Quality

We next aimed to assess the embryo quality of blastocysts derived from both IVM conditions. To that end, we evaluated the hatching rate, total cell number per blastocyst, cryopreservation tolerance, and embryo-implantation capacity.

We observed that the number of hatched blastocysts was higher in embryos that were derived from cAMP-oocytes (9.7%) compared to embryos derived from oocytes subjected to control IVM (5.7%) (Figure 6A). Interestingly, we observed that on day 7 of IVC, most cAMP-oocyte-derived blastocysts were hatching or hatched, but in control conditions, this process was not observed until day 8 of IVC. In fact, it seems as if embryos derived from cAMP-oocytes had an accelerated developmental kinetic from the beginning of IVC. On day 1 of IVC, most embryos were at the the-cell embryonic stage, and by day 6, most of them reached the blastocyst stage, whereas those derived from control oocytes mainly reached the blastocyst embryonic stage by day 7 of IVC. However, further experiments need to be conducted to validate this observation. Additionally, when the total cell number per blastocyst was determined, we observed that blastocysts derived from cAMP-oocytes exhibited a significantly higher quantity of cells than the control. Although we did not compare our in vitro-produced embryos with in vivo-derived blastocysts, it has been previously reported that bovine embryos normally have between 142 to 193 cells per embryo, similar to the number observed for the cAMP-treated blastocysts. [33]. We found that blastocysts derived from oocytes subjected to cAMP treatment had an average of 182 cells/blastocyst vs. 136 cells/blastocyst observed in blastocysts derived from control oocytes (Figure 6B). Although most of the quality parameters evaluated in the present work showed improvement in embryos derived from cAMP-oocytes, cryopreservation survival rates were similar for both groups. We observed that after thawing embryos derived from cAMP-oocytes or control conditions, the rate of expanded or alive blastocysts was similar between both experimental groups (*p* > 0.05, Figure 6C), indicating that even though treating oocytes with cAMP might lead to better quality, it does not improve blastocyst tolerance to cryopreservation.

Finally, we aimed to study the implantation rate of embryos derived from cAMP-oocytes or untreated oocytes. Two embryos at day 7-IVC from each condition from the same IVP session were transferred to surrogate mothers, using two recipients per condition, and the pregnancy rate was determined after 60 days. Embryos derived from cAMP treatment resulted in two pregnancies (one pregnancy per recipient), whereas one pregnancy was established from control conditions. Due to the low number of recipients used, further experiments need to be conducted in order to analyse if these differences can be considered significant.

In all, these results indicate that treating oocytes with cAMP modulators during IVM can lead to a higher quantity of embryos with an improved quality that is capable of establishing pregnancies.

## 4. Discussion

In vitro oocyte maturation is critical for the acquisition of full developmental competence. To induce a more physiological maturation process, biphasic IVM protocols have been proposed by a combination of an initial inhibitory culture period, using various cAMP modulators, followed by the conventional IVM phase that includes hormone supplementation [34,35]. In 2010, Albuz and colleagues proposed the SPOM system, which is based on a two-step protocol and includes cAMP modulators before and during IVM [23]. Since then, several groups have tried to replicate the results with several degrees of success. There are many factors that can influence the outcome of this protocol, and, as reviewed in Leal, G.R. et al. [36], there are differences between researchers in critical parameters such as oxygen tension; the origin and concentration of FSH used; how many hours the oocytes are matured; the BSA concentration; and if a different source of protein is being used, such as FBS. In spite of this, the original system proposal is still considered promising, which encourages adaptations that allow its application in IVP routines.

The current work investigated, for the first time, the effects of IVM systems using cAMP modulators on oocyte cytoplasmic maturation and its impact on bovine blastocyst in vitro production. Oocyte competence was evaluated both at the nuclear and cytoplasmic levels, analysing the number of oocytes that successfully reassumed meiosis, assessing the mRNA expression profile of stress-related genes, and studying the mitochondrial distribution and activity. Furthermore, we analysed the impact of SPOM-matured oocytes on blastocyst rate production and embryo quality.

As previously reported by Bernal-Ulloa and colleagues [32] and Albuz and colleagues [23], we did not find any differences in the nuclear maturation rate for cAMP-treated or untreated oocytes, indicating that the regulation of cAMP levels during IVM does not affect meiosis resumption. In contrast to the original work of Albuz et al., we employed a total maturation protocol of 24 h (2 h of preincubation followed by 22 h of maturation); however, we did not observe that shortening maturation from 26 to 24 h in total affected nuclear maturation. However, cAMP-treated oocytes exhibited differences in cytoplasmic maturation when compared with untreated oocytes. In the present study, we studied the mitochondrial distribution at different key time points during oocyte maturation that approximately corresponds to different chromatin stages: 0 h (prophase I, the germinal vesicle stage), 7 h (germinal vesicle breakdown), 15 h (meiosis I), and 22 h [37]. Our results showed that both cAMP-treated and untreated oocytes presented similar mitochondrial distribution changes during IVM, where the organelles migrate from peripheral localisation towards a more central localisation along with maturation progression. This redistribution pattern correlates with numerous reports where mitochondria were seen to disperse over the cytoplasm along maturation [38]. In fact, asymmetric mitochondrial distribution has been implicated in low oocyte competence and in future cell division arrest, whereas a proper mitochondrial distribution has been correlated with successful embryo development [39]. Thus, studying mitochondrial behaviour during oocyte maturation is relevant, as it can be considered a marker of oocyte competence and a useful embryo development predictor. It is important to note that unlike the patterns observed in other studies [40,41], the mature oocytes studied here did not show a totally diffuse pattern of mitochondria regardless of treatment. This could be indicative of suboptimal cytoplasmic maturation or simply be a product of how the samples were treated at the time of evaluation. We have observed that the use of paraformaldehyde can affect the diffusion of the Mitotracker, and that is why we decided to evaluate the unfixed samples.

Besides their localisation, the number of active mitochondria during oocyte maturation is crucial. Optimal energy production is required for oocyte and embryo development [42]. Several IVM culture media supplements have been proposed in order to enhance the mitochondrial function of in vitro matured oocytes [15]. Our results showed that oocytes treated with cAMP modulators exhibit increased mitochondrial activity in comparison to untreated oocytes. Although mitochondrial activity levels increased similarly in both groups by the end of maturation, cAMP-oocytes reached maximum activity at 7 h (coincident with the germinal vesicle breakdown stage), whilst control oocytes reached maximum mitochondrial activity at 15 h post-IVM initiation. As presented by Jeseta and colleagues, maximum mitochondrial activity patterns differ between healthy, more competent oocytes and healthy but less competent ones [37]. Mitochondrial functionality has been considered a hallmark of oocyte quality and developmental potential (reviewed by [43,44,45,46]). Our results show that cAMP-treated oocytes exhibit higher competence than the untreated ones and that this effect may be related to an early peak in mitochondrial activity during maturation. However, it would be interesting to study the oxidative levels in cAMP-treated oocytes since Gutnisky et al. reported a high positive correlation between oxidative activity and the mitochondrial activity of oocytes during maturation, indicating that an early burst in mitochondrial activity may have a detrimental effect on future blastocyst production [47].

We next investigated if the SPOM system also had an effect on the global stress level of oocytes. However, no significant differences were found between treated and untreated oocytes for any of the genes considered in this analysis. This suggests that cAMP treatment might not ameliorate nor enhance cellular stress. It is important to note that in the present work, only a small set of stress-related genes was used and that it would be interesting to evaluate the effect on cellular stress by including a wider selection of candidate genes.

Increased competence in cAMP-treated oocytes was validated by in vitro embryo production. Although the cleavage rate was similar in treated and untreated groups (83.74% and 83.76%, respectively), the number of embryos obtained using pre-treated oocytes was higher than in control conditions. These results are similar to those reported by Albuz et. al., where a higher blastocyst formation from cAMP-derived oocytes was also described [23]. However, it is necessary to mention that the blastocyst rate obtained in this study was lower than rates reported in the literature, independently of whether they were derived from cAMP or control conditions. One possible explanation for this could be related to polyspermy, a factor that is detrimental to embryo development due to genetic imbalance [48]. Usually, insemination doses are around 0.5 to 4 × 10^6^ sperm/mL, where this variation could be related to the concentration and molecule used to induce sperm capacitation as well as to the sperm quality of the bull [49]. Since we did not check polyspermy during this study, it is possible that sperm and capacitation reagent concentrations used during this study led to a high level of polyspermy, which could explain our low blastocyst development. However, it is important to remark that even if polyspermy was happening, the SPOM system significantly improved blastocyst production compared to control conditions. The prevention of polyspermy is related to proper cytoplasmic maturation, where the precise migration of cortical granules and zona pellucida hardening is required, processes that were prone to aberrancies during standard IVM conditions [50]. Future studies will assess if the SPOM system enhances the cortical granule distribution and zona pellucida modification.

Once we noticed that the SPOM system resulted in a higher number of blastocysts compared to control conditions, we next aimed to investigate the quality of the obtained blastocysts. For that, we evaluated the hatching rates, total cell number, and cryopreservation tolerance [51,52,53]. Contrary to what was described in sheep blastocysts obtained from cAMP-derived oocytes, bovine blastocysts derived from cAMP-treated oocytes resulted in significantly higher hatching rates than control conditions, indicating that this quality parameter was improved by the use of the SPOM system. Additionally, the present study revealed that the total cell number in blastocysts derived from cAMP-oocytes was higher than in control blastocysts and similar to the number reported by Albuz and colleagues [23]. It would also be interesting to study how the SPOM system affects (or does not affect) cell allocation, modifying the inner cell mass/trophectoderm proportion, as this is an important parameter for embryo viability.

The only quality parameter that showed no improvement in blastocysts derived from cAMP-treated oocytes was the cryopreservation tolerance since both groups analysed presented similar post-warming survival rates after 24 h (*p* > 0.05), indicating that further study is needed to improve this parameter.

## 5. Conclusions

All the data presented in this work suggest that SPOM- IVM’s beneficial effect on IVP could be a result of enabling the coordination between nuclear and cytoplasmic maturation, particularly accelerating the latter. To our knowledge, this is the first report to describe the effect of cAMP modulation during the in vitro maturation of bovine oocytes on mitochondrial distribution and activity patterns. We considered that factors such as the IVM media composition, cAMP modulator molecules, IVM time, and follicular size, among others, can affect the reproducibility of the SPOM protocol. Although further studies are needed to elucidate the controversial results of using cAMP modulators in cattle, we conclude that in the IVP protocol presented during this work, we found a beneficial effect of the SPOM system on embryo production and the quality parameters evaluated.

## Figures and Tables

**Figure 1 animals-14-01893-f001:**
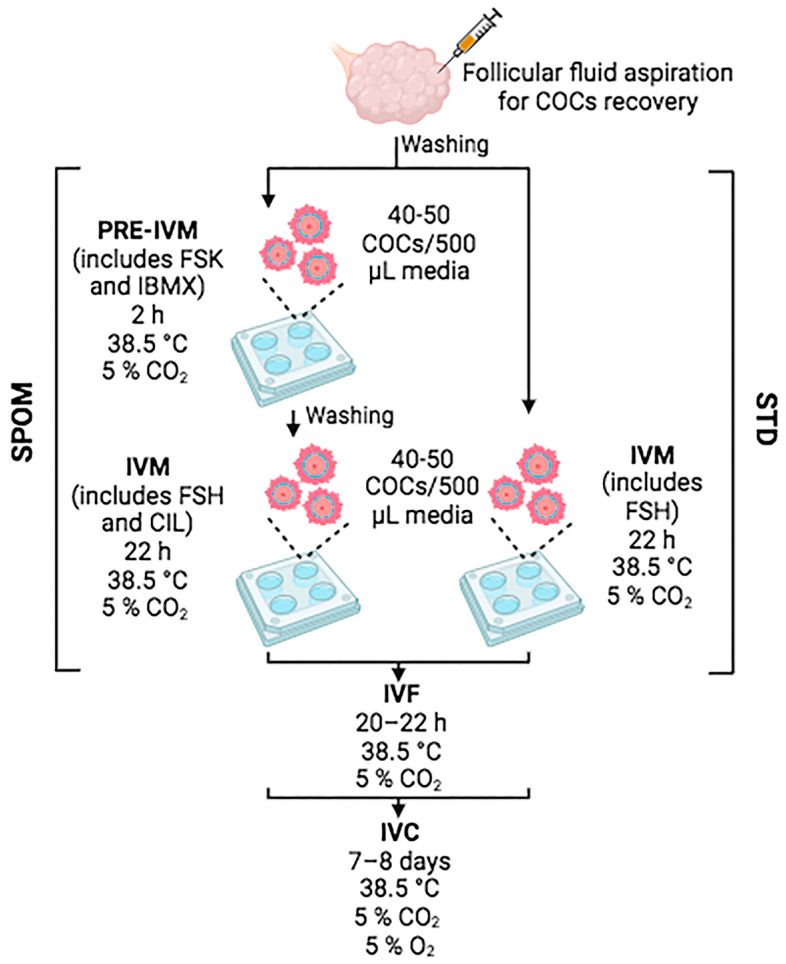
Illustration of the experimental design. Figure was created in BioRender.com.

**Figure 2 animals-14-01893-f002:**
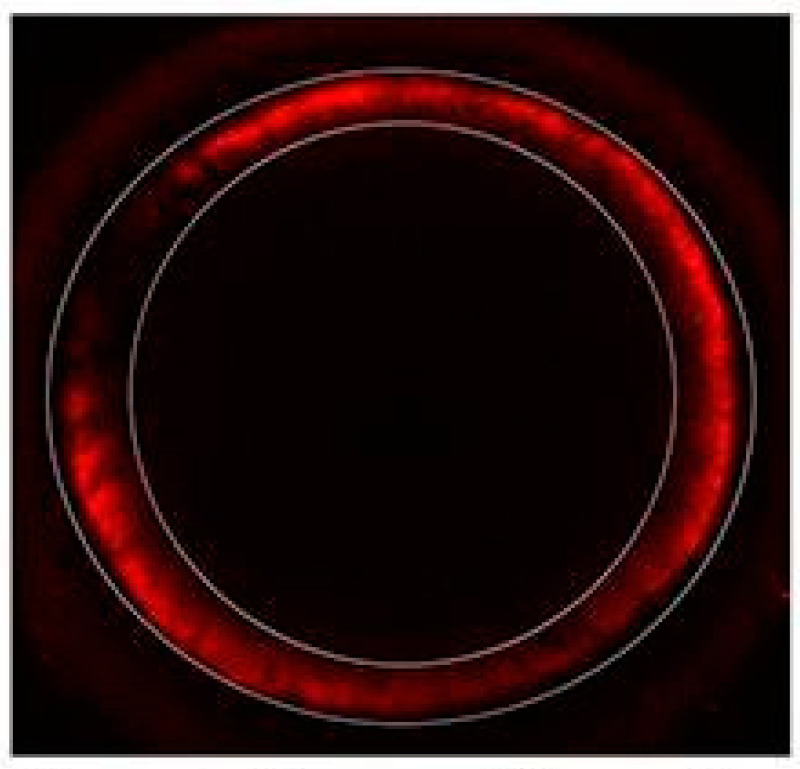
Representative confocal image showing how the area was selected in each oocyte to measure fluorescent intensity.

**Figure 3 animals-14-01893-f003:**
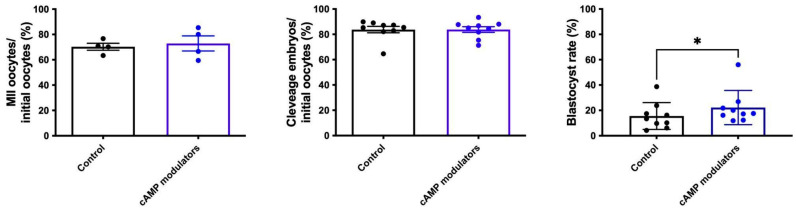
In vitro production of bovine embryos following standard IVM or IVM in the presence of cAMP modulators. Maturation rate was calculated considering the number of oocytes that showed the presence of the first polar body over the total number of oocytes subjected to maturation (*n* = 4, a total of 306 and 232 oocytes were analysed for control or cAMP modulators condition, respectively). Cleavage rate was calculated by determining the number of 2/4-cells embryos over initial oocytes (*n* = 9; a total of 729 and 634 oocytes were analysed for control or cAMP modulators condition, respectively). Blastocyst rate was determined by considering the total number of blastocysts over initial oocytes (*n* = 9; a total of 729 and 634 oocytes were analysed for control or cAMP modulators condition, respectively). * indicates significant differences (Wilcoxon test was performed, *p* < 0.05).

**Figure 4 animals-14-01893-f004:**
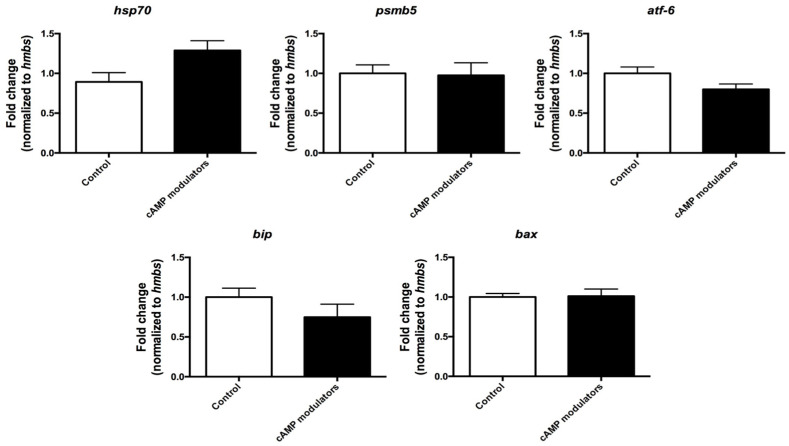
Relative quantification of stress-related genes in mature oocytes. Relative expression of *hsp70*, *psmb5*, *atf-6*, *bip*, and *bax* genes was analysed in oocytes subjected to 22 h maturation under standard conditions (control) or under the use of cAMP modulators. Expression of each gene was calculated using the standard curve method and normalised to *hmbs* housekeeping gene. Data were collected from three independent replicates from pools of 10 matured oocytes. Different letters indicate significant differences (*t*-test, *p* < 0.05). Graphic shows the fold change normalised to the housekeeping gene ± SEM.

**Figure 5 animals-14-01893-f005:**
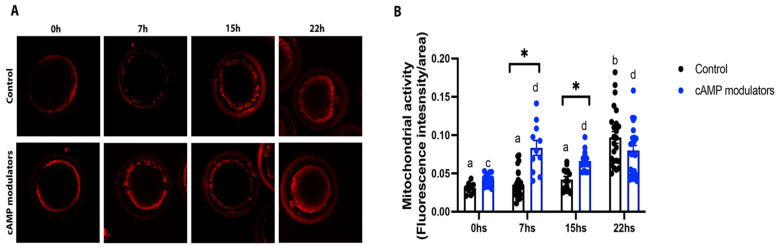
Evaluation of mitochondrial activity and distribution during oocyte maturation. (**A**) Confocal microscopy images showing mitochondrial distribution in an oocyte at 0, 7, 15, and 22 h of maturation in standard IVM (control) or in IVM with the presence of cAMP modulators. (**B**) Mitochondrial activity at 0, 7, 15, and 22 h of in vitro maturation in control conditions or under the use of cAMP modulators. Alive oocytes were stained with Mitotracker Red CMXROS and imaged using confocal microscopy. Mitochondrial activity was measured in each oocyte derived from each treatment at different maturation times by determining fluorescence intensity. Different letters in each group and * between groups indicate significant differences (two-way ANOVA and Bonferroni’s multiple comparisons test, *p* < 0.05). Data were collected from 3 independent IVM procedures, where at least 10 oocytes were imaged at each time point. Graphic shows the mean of mitochondrial activity ± SEM.

**Figure 6 animals-14-01893-f006:**
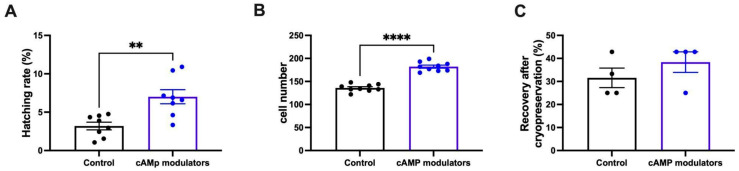
Determination of embryo quality of blastocysts derived from standard IVM or IVM with cAMP modulators. (**A**) Hatching rate of embryos that derived from standard IVM or IVM with cAMP modulators. Hatching rate was calculated considering hatched embryos over the initial number of oocytes. Data were collected after 9 independent IVP procedures. ** indicates significant differences (Wilcoxon test, *p* < 0.05). Graphic shows the hatching rate mean ± SEM. (**B**) Day 7 blastocysts derived from control IVM or IVM with cAMP modulators were stained with Hoescht 33,342 and imaged to determine total cell number. A total of 3 blastocysts from 3 independent IVP procedures (total of 9 blastocysts per experimental group) were stained to determine total cell number. **** indicates significant differences (*t*-test, *p* < 0.05). Graphic shows the mean of total cell number per blastocyst ± SEM. (**C**) Day 7 blastocysts derived from control IVM or IVM with cAMP modulators were cryopreserved and thawed for evaluating cryotolerance. Embryo recovery and expansion were evaluated after 24 h of IVC. Between 30 and 50 embryos in total from 5 independent IVP procedures were cryopreserved and thawed to evaluate cryotolerance. Graphic shows the mean of recovery rate ± SEM (ns; Wilcoxon test).

**Table 1 animals-14-01893-t001:** List of stress-related genes primers used for qPCR.

Gene Name	Forward Primer Sequence (5′-3′)	Reverse Primer Sequence (5′-3′)	Amplicon Length (bp)
70 kDa heat shock protein (*hsp70)*	*CCATCTTTTGTCAGTTTCTTTTTGTAGTA*	*GGAAGTAAACAGAAACGGGTGAA*	75
Proteasome subunit β5 *(psmβ5)*	*GCTTCTGGGAGAGGCTGTTG*	*CGGAGATGCGTTCCTTGTTT*	69
Activating transcription factor 6 *(atf-6)*	AAGCTGCTCCATTTCTCACCAT	CTTGCCTTGTGTCAGCTTCAGT	65
Immunoglobulin heavy chain binding protein *(bip)*	GATTGAAGTCACCTTTGAGATAGATGTG	GATCTTATTTTTGTTGCCTGTACCTTT	85
*Bcl2* associated X protein (*bax)*	*GCGCATCGGAGATGAATTG*	*CCACAGCTGCGATCATCCT*	59
Hydroxymethylbilane synthase (*hmbs)*	CTTTGGAGAGGAATGAAGTGG	AATGGTGAAGCCAGGAGGAA	80

## Data Availability

The original contributions presented in the study are included in the article, further inquiries can be directed to the corresponding author.

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
