# Peer review of "The Simulated Physiological Oocyte Maturation (SPOM) System Enhances Cytoplasmic Maturation and Oocyte Competence in Cattle"

_animals, 2024, doi:10.3390/ani14131893_

Round 1

Reviewer 1 Report

Comments and Suggestions for Authors

Line 124 … mention approval number

Line 125 – How was size of follicles measured?

Line 127 :Standard IVM conditions consisted of incubating groups of 40-50 COCs… include Ref

Line 199: How was the house keeping gene selected?

Reviewer 2 Report

Comments and Suggestions for Authors

I am reviewing the manuscript entitled 'The SPOM system enhances cytoplasmic maturation and oocyte competence in cattle.' for consideration to be published in Animals. In this manuscript, the SPOM system was applied to cattle oocyte maturation in vitro. The nuclear and cytoplasmic maturation and embryo quality were evaluated by oocyte maturation rate, cleavage rate, blastocyst rate, mitochondrial activity, hatching rate, total number of cells per blastocyst, cryopreservation tolerance and embryo implantation ability. The results showed that the use of cAMP modulators during IVM in IVP system improves cytoplasmic maturation, embryo quality. The experimental design of the manuscript is reasonable. There are some major points to be improved on this manuscript:

1. The abstract description is too lengthy.

2. The maturation rate, cleavage rate, Blastocyst rate in Table 2 should be written as 72.05%a , 76.20%a.

3. Description in line233, the manuscript studied the effects of cAMP modulators on cytoplasmic maturation by evaluating stress-related genes and mitochondrial activity, but the results were contrary and were not discussed.

4. The manuscript did not describe the method of embryo implantation or pregnancy rate.

5. Figure 3B and 3C shows no letters      

Reviewer 3 Report

Comments and Suggestions for Authors

Comments to the author:

The manuscript “The SPOM system enhances cytoplasmic maturation and oocyte competence in cattle” aimed to evaluate the effect of regulating cytoplasmatic maturation on blastocyst production rate and quality. The study was well presented and written. The introduction defends the problematization of the study, informing the need for improvement in the in vitro cultivation of oocytes. An important point of the study is the authors discussing whether the rates obtained (IVM, cleavage and blastocysts) were similar or higher than those in the literature. It is interesting for the authors to discuss this point, especially because the system used by the authors resulted in rates of approximately 70%. Furthermore, the methodology is clear, and an experimental design topic is important to facilitate understanding of the problematization. Results were well presented. Other specific comments are presented for each section:

Title, summary, and abstract:

1. Define the SPOM acronym in the manuscript title.

2. Delete “…and prevent sexual diseases.”

3. Shorten the abstract introduction. It seems too long.

4. In abstract, define IVP and cAMP.

5. In abstract, specify the phrase "...were able to establish pregnancies."

Introduction:

1. The second paragraph of the introduction seems too long. I suggest splitting them and re-arranging it.

2. Explain in general that other factors may affect embryonic competence in IVP.

Material and methods:

1. Why did they use recombinant FSH in the IVM medium?

2. I suggest that authors present the experimental design of the manuscript as a material and methods topic.

Results and discussion:

1. All tables cannot be figures, but rather a table.

2. Explore better the results of pregnancy rates, success percentage...

3. Discuss further why modulators were not effective in the number of blastocysts produced. What factors may be involved throughout the system?

4. The IVM, cleavage and blastocyst rates obtained in the present study were similar to or higher than the literature average? It is interesting for the authors to discuss this point, especially because the system used by the authors resulted in rates of approximately 70%.

Reviewer 4 Report

Comments and Suggestions for Authors

The authors explored the use of the Simulated Physiological Oocyte Maturation (SPOM) system to try to improve bovine oocyte competence during in vitro maturation and in turn blastocysts production. The authors evaluated mitochondria activity and gene expression on oocytes exposed to the SPOM system. The authors also evaluated the resultant embryos in terms of cryopreservation tolerance and total cell number. In principle, the paper provides some novel information (mitochondria activity in oocytes), but it needs clarification in some sections regarding the methodology used to generate data. Moreover, some statements are not backed up with statistical analysis and some analyses are not robust enough to present as evidence of beneficial effects of the SPOM system.

These are my recommendations:

1)Section 2.1.: Please provide details of the medium used for collection of oocytes and indicate for how long the oocytes were outside the follicle before being placed in the SPOM system.

2)Section 2.2.: The authors used a concentration of 2x106 sperm/ml in their IVF protocol. This is double than what it is usually used, can you provide the level of polyspermy you are having using this sperm concentration? If not, discussion of a possible increase in polyspermy levels is necessary, particularly in relation to the blastocyst production results reported in your control group.

-Please provide details of the protocol use for embryo recipient synchronisation, the procedures used for embryo transfer, and information on ethical approval for the use of live animals in this research. This is relevant to demonstrating ethical used of animals in this paper. 

3)Section 2.6.: Please provide more detail about the protocol used for the analysis of mitochondria in oocytes. For example, what media was used when the mitotracker solution was added, and how were embryos mounted on microscope slides for confocal analysis? (or were they kept in medium during analysis?). Provide the size of Z-stacks in µm. Authors need to provide more detail of how they performed the fluorescence measurements, i.e., what areas of the oocyte were exactly used for measurements, please provide a figure illustrating this (See figure 3 in this paper: Asimaki et al. 2022. Front. Toxicol. 4:811285. doi: 10.3389/ftox.2022.811285). This is necessary due to the presence of autofluorescence observed in the zona pellucida from the images. Also, it is not clear whether the authors used a 3D reconstruction to measure fluorescence levels or if they use a 2D picture from a middle confocal plane. Please clarify. Importantly, how did you correct your data for oocyte size?

-All this is essential to report in your paper to improve reproducibility of these data.

4)Section 2.7.: Please provide the test you used to check normal distribution in your data. Usually arcsine transformations are needed to analyse percentage data as a continuous variable (the authors used T-test and ANOVA to analyse their data).

5)Section 3.1:

-Lines 226-232: The presentation of data on table 2 needs to be improved. My strong recommendation is to present these data using a combination of bar charts and scatterplots. See rationale here:

Vail A, and Wilkinson J. 2020. Bang goes the donator plot! REPRODUCTION, 159(2):E3-E4.

from https://rep.bioscientifica.com/view/journals/rep/159/2/REP-19-0547.xml

Weissgerber TL, et al. 2015. Beyond bar and line Graphs: Time for a new data presentation paradigm. PLOS BIOLOGY, 13(4): e1002128. https://doi.org/10.1371/journal.pbio.1002128

 Weissgerber TL, et al. 2019. Reveal, don’t conceal. CIRCULATION, 140:1506-1518.

https://www.ahajournals.org/doi/10.1161/CIRCULATIONAHA.118.037777

 This is feasible since the authors treated percentages as continuous variables (i.e., they used T-test). This will allow for clearer visualization of the data variation, this is, the mean production per replicate for each group across the three variables reported (maturation rate, cleavage rate and blastocyst production).

-Importantly, the authors need to run their analysis on blastocyst production using the number of oocytes matured. They can keep the blastocyst production based on cleaved embryos, but blastocyst production based on the number of oocytes matured must be included in the results section as well (section 3.2). This will give a clear indication of the robustness of authors’ in vitro embryo production model. This is important, because the authors are reporting a blastocyst rate of 21.4% in their control group based on the number of cleaved embryos. This means that their production based on number of oocytes cultured is way below the worldwide range 20-40 % (blastocysts/oocytes used, see Ferré et al, 2020 Animal 14[5]:991-1004; Velazquez 2023 Veterinary Sciences 10[10]:604). And this is more relevant to present given that the treatment was applied on the oocytes.

-The data reported in table 2 are a bit confusing in the “Standard IVM” group as the number of oocytes used to evaluate maturation rate (441) is lower than the number of cleaved embryos (461). Please correct as appropriate.

-The “n=” is confusing the way it is written in the legend of table 2. I am assuming it makes reference to replicates rather than the number of oocytes and cleaved embryos used for calculation of production rates. Please use “n=” to indicate the number of structures used in your experiments and clearly indicate the number of replicates used.

-Lines 254-256: The authors stated: “We observed that along maturation progression, mitochondria migrated from the peripheral region to the centre of the oocytes in both groups (Figure 2A).” But this is not what the figures are showing. The fluorescence is clearly retained at the periphery of the oocytes, the centre of the oocyte remained free of fluorescence in all the figures provided by the authors. All you can say is that there was an increase in fluorescence intensity. Please rephrase as appropriate

-Data in figure 2B should also be presented using a combination of bar charts and scatterplots (see rationale above).

6)Section 3.2.: As indicated above, the results in this section should be presented using the analysis based on the number of oocytes cultured to back up the statement in lines 289-291.

7)Section 3.3.: Hatching rates need to be calculated based on the number of oocytes matured. The treatment tested by authors was during oocyte maturation, and as such this is the experimental unit, not the blastocyst produced in your system. Please provide these data and rephrase your discussion as appropriate.

-Data in figure 3 should also be presented using a combination of bar charts and scatterplots (see rationale above). In the legend of figure 3 the authors indicated: “Three blastocysts from 3 independent IVP procedures were stained for determining total cell number”. Are you saying that 3 blastocysts were stained per treatment? If so, these data are not robust enough to determine the effect of treatment on embryo cell number, and should not be presented in the paper. At least 10 embryos per treatment should have been analysed.

-Lines 299-305. The authors need to clearly indicate that the description of embryo kinetics is NOT significant, since they do not analyse these data with statistics. This is important to clarify in the paper otherwise it is misleading information.

-Lines 305-310: The authors wrote: blastocysts derived from cAMP-oocytes not only exhibited a significantly higher quantity of cells but also showed a total cell number that resembled in vivo cattle embryos”. The statement is misleading since the difference in cell number was not significant according to the information presented in figure 3. But most importantly, the use of 3 embryos per group is not suitable to make such statement. Also, the comparison with in vivo embryos is not appropriate since the authors did not analyse in vivo produced embryos. Making comparisons from data in the literature is not a scientifically sound approach to address this topic, especially with the very low number of embryos used in your study (3 per group). Please delete this statement.

-Lines 332-338: It is not clear how many embryo recipients were used per treatment, please provide this information. The authors mentioned that two embryos were transferred per recipient, please indicate whether both embryos transferred were blastocysts, or if a combination of morulae and blastocysts was used. Were there any twin pregnancies in these embryo transfers? If so, were there any complications with the embryo recipients? Do you have data on calving rate at this time? Please provide this information if available.

8)Section 4: It will be very useful if the authors could briefly discuss the reason for choosing 2 hours of treatment (other than just following a protocol from previous papers). Could a longer exposure to this treatment may become detrimental?

-Lines 364-367: the authors wrote: “In the present study, we studied mitochondrial distribution at different key time points during oocyte maturation: 0 h (prophase I, germinal vesicle stage), 7 h (germinal vesicle breakdown), 15 h (meiosis I) and 22 h (metaphase II and polar body extrusion).” However, the only way these different stages could have been monitored is if the authors had used a DNA dye to determine chromatin stages (especially before polar extrusion), which the authors did not. As such, this statement is not true and should be rephrased to indicate that the hours chosen to analyse mitochondria activity roughly correspond to different chromatin stages. Importantly, the authors need to provide a reference that backs up their timing of chromatin configuration during IVM in cattle.

-The authors must discuss their results on blastocysts production based on the analysis using the number of oocytes cultured, not on the number of cleaved embryos (e.g., statement in lines 398-400 might need to be changed). Importantly, the authors need to discuss the absence (in their figures) of a diffused mitochondrial distribution expected to be seen in mature bovine oocytes.

See figure 9 in this paper: Feng et al. 2024. Int. J. Mol. Sci. 25(9), 4808; https://doi.org/10.3390/ijms25094808

See figure 2 in this paper: Asimaki et al. 2022. Front. Toxicol. 4:811285. doi: 10.3389/ftox.2022.811285

This suggests that although the SPOM system used in this study appears to increase the presence of active mitochondria, it does not reach levels indicative of good oocyte maturation. This is reflected in the low blastocyst production observed in this study (which appears to be lower than 20% based on the number of matured oocytes). My point is, there are IVF labs achieving over 30% blastocyst production from matured oocytes. Do the authors believe that the SPOM system will further improve blastocyst production in these systems? I hope the authors can discuss the practical implications of this and provide a more suitable conclusion at the end of the paper.

-Lines 380-382: The authors stated: “Our results showed that oocytes treated with cAMP modulators exhibit an accelerated kinetics in mitochondrial activity in comparison to untreated oocytes”. Please remove this sentence, as it is misleading since your data analysis does not support this statement.

-Lines 401-403: The authors wrote: “these embryos showed increased developmental kinetics since most of them reached the 2-cells embryonic stage on day 1 of IVC and hatched by day 7”. This statement is misleading as it is not backed up with a statistical analysis. Please remove it.

-Lines 404-406: The authors stated: “the present study revealed that total cell number in blastocysts derived from cAMP-oocytes not only was higher than control blastocysts, but also similar to in vivo produced embryos”. This statement is not based on robust data (please see comments in item 7 for lines 305-310) and should not be included. Please remove it.

Round 2

Reviewer 4 Report

Comments and Suggestions for Authors

I thank the authors for addressing some of my recommendations. However, there are some important issues that still need to be addressed.

1)The authors replied: “The details of the protocol used for embryo recipient synchronisation and the procedures used for embryo transfer was included in section 2.2. Ethical approval documents for embryo transfer were included in the paper submission (FILE NBR 0034/22)”.

However, the ethical approval concerns only the use of ovaries from the abattoir. There is no ethical approval for the use of animals (embryo recipients) for embryo transfer. This needs to be provided if the authors want to include the embryo transfer data in their paper.

2)Regarding the measurement of mitochondria, the authors indicate in the paper (lines 227-229): “For analysis, we made a projection of the summarised Z stack in one plane. The fluorescence intensity was measured in the selected area (see figure 2)”.

The phrasing is confusing, as “projection of the summarised Z stack” suggest a 3D reconstruction of the oocyte, but the figure presented is representative of an equatorial region. Indeed, the authors later indicated this when replying another comment: “…here the images represent a section of the equatorial region of the oocyte..”.

So, please indicate clearly that a single equatorial region was used for the measurement of fluorescence.

Also, I am assuming that the “selected area” is the “ring” where fluorescence is present, rather than the inner circle. But the authors, in their first submission, indicated that “…along maturation progression, mitochondria migrated from the peripheral region to the centre of the oocytes in both groups”. This will indicate that fluorescence was measured by the authors in the whole of the cytoplasm to begin with, and I was expecting that perhaps the authors will mention a similar methodology described in the paper they suggest for discussion of their mitochondrial staining pattern (Gutnisky et al. 2014, Reprod. Fertil. Dev. 26[7]:931-942). But now the authors indicate that it was in a “ring-like” specific area of the cytoplasm (as shown in their figure) that measurements were done. If the authors want to change the way they analysed fluorescence to match what it is presented in the reference I suggested (I just did it so that they could see an example), then they need to reanalyse the oocytes with this new “protocol”. On this note, the authors will need to ensure that the area of the “ring” is of the same size for every single oocyte they analyse.

Also, in the new version the following information is provided: “Active oocyte’s mitochondria were stained using 100 nM Mitotracker Red CMXROS (Molecular Probes) prepared in Dulbecco's Phosphate-Buffered Saline (DPBS) supplemented with 3 mg/mL for 30 min at 38.5 °C.”

It is not clear what the “3 mg/ml” refers to. Please clarify this.

3)The authors replied: “The pattern observed in our images resembles the ones observed in different publications using the same Mito tracker as we employed in our work (ISSN: 0006-3363. http://www.biolreprod.org; DOI: 10.1071/RD12397). Is worth noting that staining can be influenced depending whether the oocyte is fixed or in vivo stained and if oocytes are in vitro or in vivo matured. Also, here the images represent a section of the equatorial region of the oocyte whether others present a 3D recreation of the z- stack.”

The paper mentioned by the authors (Gutnisky et al. 2014, Reprod. Fertil. Dev. 26[7]:931-942) used MitoTracker Green, whereas the authors utilised Mitotracker Red CMXROS in this paper. Several papers working with Mitotracker Red CMXROS have documented a peripheral or diffused pattern of mitochondria in cattle oocytes:

Somfai et al. 2015. Anim. Sci. J. 86: 970-980. https://doi.org/10.1111/asj.12387

De Oliveira et al. 2022. Anim. Prod. Sci. 62(2):142-141 https://doi.org/10.1071/AN21197

Boruszewska et al. 2020. Reprod. Biol. Endocrinol. 18:40 https://doi.org/10.1186/s12958-020-00598-9

Niu et al. 2020. J. Anim. Reprod. Biotech. 35(1):102-111 https://doi.org/10.12750/JARB.35.1.102

Importantly, when comparing in vivo vs in vitro matured oocytes the peripheral pattern increased in in vitro matured oocytes, suggesting that a peripheral pattern is indicative of impair maturation (Somfai et al. 2015). So, the fact that no disperse mitochondrial was detected in any of the oocytes analysed in the model used by the authors need to be discussed in the paper. But assuming that there is no difference between MitoTracker Green and Mitotracker Red CMXROS to analyse mitochondrial activity in cattle oocytes, the authors need to discuss why the increase in mitochondrial activity in the control group at 15 and 22 hours during IVM reported by Gutnisky et al. 2014 (which has virtually the same measurement time points: 0,7,15,22 vs 0,9,15,22) was not replicated in the present study (i.e., in control groups).

On this note, the authors used Jeseta et al. 2014 to compare their data (lines 431-435): “These results are in accordance with the ones presented by Jaseta and colleagues, where they reported that healthy more competent oocytes significantly increased their mitochondrial activity before resumption of meiosis while healthy but less competent oocytes increased their mitochondrial activity significantly before completion of maturation [40].”

In the paper of Jeseta et al. 2014 they (Jeseta et al) referred to their data on “frequency of oocytes with clusters” (table 3a in that paper) to make their inaccurate assumption as the data on “mean mitochondrial activity per oocyte” they reported was ignored in their discussion (i.e., Jeseta ignored that data). As such, the mean mitochondrial activity reported by Jeseta (Figure 2a in that paper reporting “mean mitochondrial activity per oocyte”) indicates that mitochondrial activity increases at 7 hours (in “healthy” oocytes from medium follicles) when compared with 3 and 16 hours during a 24-hour IVM protocol, which does not match either the control or the “cAMP modulators” group in the present paper. Hence, that needs to be removed from your discussion (i.e., lines 431-435) and perhaps try to put in perspective the different patterns of mitochondrial activity reported in previous papers especially the ones from Jeseta et al 2014 (but using the “mean mitochondrial activity per oocyte” reported in that paper) and Gutnisky et al. 2014.

4)Assuming the authors clarify how exactly they quantified fluorescence levels (see comment 2), their control data is basically different from previous papers, and this needs to be highlighted and explained in their discussion. Nevertheless, it seems that an early increase in mitochondrial activity seems to lead to an increased cell number in the resultant blastocysts. The authors reported that the SPOM treatment increased blastocyst production. However, using the data provided by the authors (there are only 8 replicates for blastocyst rate, not 9 as the authors indicated in the paper) I run a Wilcoxon rank sum test (Mann-Whitney in SPSS), and the statistical outcome indicate that there is no significant difference between the control and the SPOM group:

Mann-Whitney Test

Ranks

Group

N

Mean Rank

Sum of Ranks

BlastPROD

Control

8

6.88

55.00

SPOM

8

10.13

81.00

Total

16

Test Statisticsa

BlastPROD

Mann-Whitney U

19.000

Wilcoxon W

55.000

Z

-1.365

Asymp. Sig. (2-tailed)

.172

Exact Sig. [2*(1-tailed Sig.)]

.195b

a. Grouping Variable: Group

b. Not corrected for ties.

And the lack of effect of SPOM is evident in the data distribution shown infigure 3, where the mean production and SEM overlap significantly.

Furthermore, I run a T-test for the hatching and cell number, and the statistical outcome showed that hatching was not affected:

T-Test

Group Statistics

Group3

N

Mean

Std. Deviation

Std. Error Mean

Hatching

Control

9

.2261

.20499

.06833

SPOM

9

.3174

.18145

.06048

Independent Samples Effect Sizes

Standardizera

Point Estimate

95% Confidence Interval

Lower

Upper

Hatching

Cohen's d

.19358

-.472

-1.403

.474

Hedges' correction

.20328

-.449

-1.336

.451

Glass's delta

.18145

-.503

-1.444

.467

a. The denominator used in estimating the effect sizes.

Cohen's d uses the pooled standard deviation.

Hedges' correction uses the pooled standard deviation, plus a correction factor.

Glass's delta uses the sample standard deviation of the control (i.e., the second) group.

 This is the input used in SPSS based on the data sent by the authors:

Therefore, neither blastocyst production nor hatching rate appear to be improved by SPOM, at least based on the data sent by the authors. Please rectify this in your paper and discuss it accordingly.

As the authors highlighted in their discussion, Albuz et al 2010 claim an increase in blastocysts production associated with the SPOM protocol with a similar low blastocyst rate as in the control group of this paper (around 22% blastocyst rate in control group using blastocyst/2-cell embryos to calculate embryo production in a 24-hour IVM protocol). However, other groups with a more robust embryo production did not find any increase in blastocyst formation when using SPOM:

 -25% (control) vs 26%, Bernal-Ulloa et al. 2016, PLoS ONE 11(2): e0150264 https://doi.org/10.1371/journal.pone.0150264

-27% (control) vs 18%, Ezoe et al. 2015, PLoS ONE 10(5):  e0126801 doi:10.1371/journal.pone.0126801

Moreover, there are reports of a negative effect on blastocysts production:

-32% (control) vs 23%, Bernal et. al., 2015, Zygote 23(3):367-377. doi:10.1017/S0967199413000658.

-31% (control) vs 16%, Leal et al. 2018, Anim. Prod. Sci. 59(4):634-640 https://doi.org/10.1071/AN17895

-50% (control) vs 14%, Guimarães et al. 2015, Theriogenology 83(1):52-57 https://doi.org/10.1016/j.theriogenology.2014.07.042

On the other hand, it has been suggested that the length of IVM can make the difference as one research group reported negative effects using a 28-hour IVM protocol (Leal et al. 2018), but an increased blastocyst production was reported (25% [control] vs 33%) when a 24-hour IVM was used (Leal et al. 2020, reference 28 in this paper).

Since the authors used a 22-hour IVM protocol, they should discuss the possible implications of their IVM period as it differs from the IVM period used in Albuz et. al. 2010.

The main reason I am mentioning all this is because the authors concluded in their paper: “We consider that the SPOM-IVM system is beneficial for blastocyst production and embryo quality and thus, it can be used in IVP system routines to further increase the rate of blastocyst production”. There is insufficient data to support this claim, especially given the poor embryo production reported in this paper (and there seems to be no clear indication that blastocyst rate was improved). The authors need to consider that only a few studies have successfully increased blastocyst production using the SPOM system (see the systematic review in reference 36 in this paper), and more importantly, in some cases it was detrimental for blastocyst formation. The only conclusion the authors can make is that under the circumstances of their poor in vitro production an increased cell number was achieved with the SPOM system, presumably associated with an increased in mitochondria activity (although the authors seem to be unable to speculate how this can be possible). Even if the authors replied with new data that showed that 7% increased (15% vs 22%) is significant between the groups, the blastocyst rate achieved with the SPOM system in this paper is significantly below the current global average embryo production in cattle (around 30%). This hinders the use of these data to actively promote SPOM for increasing blastocyst production in the bovine embryo industry. Therefore, the authors do not have sufficient grounds to conclude that “it can be used in IVP system routines to further increase the rate of blastocyst production.” Please revise your conclusion accordingly.

5)The authors mentioned in the new version of their paper the following (lines 455-456): “Usually, insemination doses are around 0.5 to 4 x106 sperm/mL”. I don’t think a sperm concentration of 3-4 x106 sperm/mL is common in cattle IVF. I will advise to remove this statement.

The authors also mentioned in their paper the following: “However, it is important to remark that even if polyspermy was happening, the SPOM system significantly improved blastocyst production compared to control conditions [46]”. And they use the reference from Papi et al. 2012 to back up this statement. However, Papi et al 2012 did not analyse levels of polyspermy in a SPOM protocol. Please remove this misleading statement.

6)References 48-49 should be removed from the “References” section as they are not mentioned in the text.

7)Finally, the authors indicate that a net increase in cell number in blastocyst from the SPOM system is indicative of increased quality. However, here the authors should discuss that other important cell parameters were not evaluated, including cell allocation. This is an important parameter, as inner cell mass/trophectoderm proportion (around 25-35% of cells should be allocated to the inner cell mass) is important for embryo viability. Also, levels of apoptosis were not analysed in the present study, and there is a possibility that with increased number of cells a higher number of apoptotic cells might be present.

Comments on the Quality of English Language

No issues

Author Response

Dear reviewer, please find attached a file with our responses to your comments in blue. if you have any further question do not hesitate to contact us .

Best regards

Carolina Blüguermann, PhD
